# Mental Health Status before and during the COVID-19 Pandemic in Patients First Visiting a Psychosomatic Internal Medicine Clinic in Tokyo, Japan

**DOI:** 10.3390/ijerph19042488

**Published:** 2022-02-21

**Authors:** Fumio Shaku, Motoko Ishiburo, Masako Miwa, Shuichiro Maruoka

**Affiliations:** Department of Psychosomatic Internal Medicine, Itabashi Hospital, Nihon University, 30-1, Oyaguchikamicho, Itabashi-ku, Tokyo 1738610, Japan; ishiburo.motoko@nihon-u.ac.jp (M.I.); miwa.masako@nihon-u.ac.jp (M.M.); maruoka.shuichiro@nihon-u.ac.jp (S.M.)

**Keywords:** anxiety, COVID-19, depression, mental health, pandemic, psychosomatic, CES-D, STAI, GHQ

## Abstract

The coronavirus disease (COVID-19) pandemic has aggravated anxiety and depression worldwide, particularly in patients with chronic illnesses and mental disorders. Using validated questionnaires, in this paper, we examine the psychological effects of the pandemic in Japan in detail. The General Mental Health Scale (GHQ), the State–Trait Anxiety Inventory (STAI), and the Center for Epidemiologic Studies Depression Scale (CES-D) were used to assess mental health, state and trait anxiety, and depression, respectively. The survey was completed during the patients’ first visits to the clinic of Psychosomatic Internal Medicine from November 2018 to April 2021. The Mann–Whitney U test was used to compare data from 226 participants before and during the COVID-19 pandemic. The GHQ total, GHQ subscales of “social dysfunction” and “anxiety and dysphoria”, and state anxiety scores were significantly higher during than before the pandemic. The GHQ total, some GHQ subscales, and state anxiety scores were significantly higher among male than female participants during the pandemic. The GHQ total, some GHQ subscales, CES-D, and state anxiety scores in those aged 10–39 years were significantly higher. Thus, the COVID-19 pandemic may have caused mental health changes in many individuals based on their gender, age, and with time.

## 1. Introduction

The first coronavirus disease (COVID-19) infection in Japan was confirmed on 16 January 2020. The World Health Organization declared the virus a pandemic on 11 March 2020, and the Japanese government first declared a state of emergency on 7 April 2020 [1].

With the spread of COVID-19, social activities have been restrained in many countries, and movement is restricted depending on quarantine status. The fear of infection, behavioral restrictions, and financial implications can cause several psychological issues. A worrying consequence of the pandemic has been the increase in mental health problems, including panic symptoms, depression, stress, and anxiety [2]. Among these psychological symptoms, pervasive community anxiety and generalized fear are the most common. These symptoms are related to the COVID-19 pandemic, which has seen an increase in new patients with panic symptoms and anxiety. Several studies proposed that psychotic symptoms, anxiety, and depression are likely to worsen during extreme COVID-19-induced social disruption and stress [3]. A fear of the unknown leads to high anxiety levels in both healthy people and those with pre-existing mental health problems; unjustified public fear may lead to scapegoating, discrimination, and stigmatization [4]. Moreno et al. [5] reported that mental health problems are apparent in the general population, as well as in people with mental disorders and in frontline workers. Symptoms that include fear, depression, anxiety, sleep problems, and stress have become increasingly common during the COVID-19 pandemic [6]. The relatively high anxiety and depression levels and rates (45.1% and 23.6%, respectively) are not unexpected findings, considering the psychological influence of the pandemic. A previous Chinese study indicated that 35% of people were psychologically influenced by the pandemic [7]. Other studies reported depression rates of 16.5–48.3% in the general Chinese population during the same period, which reveals a noticeable effect of the crisis [8,9].

Pappa et al. reported that the prevalence rate of anxiety and depression appeared to be higher among females. Moreover, compared with doctors, nursing staff exhibited higher prevalence estimates for both anxiety and depression, in a study among healthcare workers [10]. The prevalence rate of depression was reported to be higher in women than in men, which could reflect the established gender gap for symptoms of depression [11].

Bobevski et al. found that women, individuals with present or past psychiatric diseases, and individuals with chronic illnesses have a greater sensitivity to and awareness of sensations in their bodies, and their health-related anxiety may be higher than that in men [12]. Özdin et al. reported [13] a high level of health-related anxiety in people with chronic illnesses, emphasizing the psychological effects of the COVID-19 pandemic. Vulnerability to and symptoms of depression and anxiety during the pandemic were reported in individuals with autism spectrum conditions [14]. In an online study, Fernández-Aranda [15] found that 56.2% of patients with eating disorders reported experiencing additional anxiety, and 37.5% reported experiencing eating disorder symptomatology during the pandemic. Based on these results, it is believed that people with psychosomatic disorders, including eating disorders, are generally more susceptible to the COVID-19 pandemic than those without.

Most mental health studies on the COVID-19 pandemic were performed in China [16], with fewer studies being conducted in the Western countries. Previous studies reported a variety of outcomes as a result of the pandemic [16].

It has been suggested that mental health issues are based on individual and cultural backgrounds. In a recent study, Ishikawa et al. reported that the prevalence of mental disorders in Japan is lower than that in the Western countries; furthermore, a greater lifetime prevalence for men and longer persistence for women seems to be a unique feature of people in Japan, which could be associated with cultural differences in gender-related etiology [17]. The prevalence of drug abuse/dependence in Japan was especially lower than in the U.S.A. and other Western countries. Greater gender discrimination in Japan may be associated with persistence of mental disorders among females [17]. Kudielka reported [18] that expressing anger is considered shameful in the Japanese culture and that men found it difficult to express anger, especially in public. Cultural and social environments have been different for men and women (e.g., life events, gender role norms at home and job positions and/or salary gaps, and biological sex differences also led to sex differences in stress reaction) [19].

At our clinic, we specialize in psychosomatic disorders; in the case of psychosomatic disorders, patients often have chronic physical illnesses. Several patients have obvious psychosocial backgrounds and may have higher sensitivity to the pandemic, and have different tendencies than those who are physically and mentally healthy. Therefore, we focused on patients who visited the clinic for psychosomatic internal medicine for the first time. The patients generally have both mental and physical problems, and few past studies have focused on patients undergoing psychosomatic medicine. The novelty lies in the fact that the study deals with a psychosomatic medicine service.

This study investigates in detail how the COVID-19 pandemic affected the patients’ mental health and hypothesized that the mental health status of the patients changed as a result of the COVID-19 pandemic.

## 2. Materials and Methods

### 2.1. Study Design

This was a quantitative study to examine the mental health of patients who visited a psychosomatic internal medicine clinic before and during the COVID-19 pandemic. This study was approved by the Human Subjects Institutional Review Board of Nihon University Itabashi Hospital Clinical Research Judging Committee (approval no. RK-210309-5). The study was conducted in accordance with the principles of the Declaration of Helsinki. Informed consent was obtained in the form of the “opt-out” option on the website. Those who opted-out were considered as unwilling to participate and were excluded from the study.

### 2.2. Participants

The pre(before)-pandemic group comprised patients who first visited the Department of Psychosomatic Internal Medicine in Nihon University Itabashi Hospital, from November 2018 to December 2019. The during-pandemic group comprised patients who first visited the department from 7 April 2020 (the date on which the Japanese government first declared a state of emergency) to April 2021. Of the 282 participants, the inclusion criterion was: all patients (18 years old or older who could perform a psychological test) who visited the Department of Psychosomatic Medicine for the first time (referred from another clinic). The exclusion criteria were: (1) patients who could not undergo a psychological test, (2) those who requested refusal from participating in this research, and (3) others who were judged by the principal investigator to be ineligible as research subjects. After eliminating patients with incomplete data or those who previously attended the clinic (i.e., it was not their first visit), we had a final total of 226 patients. Since the two non-Japanese participants were in Japan for a long time, they were treated as being close to the Japanese sense based on previous articles [20,21].

Data was collected retrospectively on the patients’ demographic characteristics and implemented routines, through anonymous questionnaires, including the General Mental Health Scale (GHQ-30), the State–Trait Anxiety Inventory (STAI), and the Center for Epidemiologic Studies Depression Scale (CES-D), at their first visit to the department.

This clinical study only used medical information and did not implement invasive procedures or interventions. Thus, informed consent was not required, in accordance with the government guidelines. Information was disclosed with respect to the research purpose, and the patients had the opportunity to refuse participation.

### 2.3. Questionnaires

#### 2.3.1. The GHQ-30

The GHQ is a self-reported measure for the screening of non-psychotic psychiatric diseases [22]. It is used in both epidemiologic studies and clinical settings to investigate the mental health of the general population. Hayashi et al. developed the Japanese version of the GHQ [23]. There are several versions of this self-administered questionnaire, with 60, 30, 28, and 12 items. The GHQ-30 is the most commonly used measure; it is a shortened version of the GHQ-60, excluding somatic items, but remains an essential measure of general psychological problems. An important characteristic of the GHQ-30 is the inclusion of an equal number of negatively and positively phrased questions that have highly similar verbal anchors for their answer categories [24].

#### 2.3.2. The STAI

Anxiety was evaluated using the STAI. This self-reported questionnaire includes two scales that assess different dimensions of anxiety (i.e., state and trait anxiety). While the State Anxiety Scale (STAI-S) assesses responses at specific moments in time, the Trait Anxiety Scale (STAI-T) determines how subjects generally feel [25]. Scores can vary from 20 to 80 points (with <30 indicating little or no anxiety, 31–49 indicating moderate anxiety, and ≥50 indicating extreme anxiety) [26].

#### 2.3.3. The CES-D

The CES-D is an established, simple self-reported measure that is used to evaluate depressive symptoms. It comprises 20 items and assesses depression symptoms from the total score of 60 (range 0–60). The results were interpreted as follows: normal (0–15) and depression (16–60) [27]. This scale (as well as the GHQ-30 and STAI) is reliable and available for the Japanese population [28].

### 2.4. Statistical Analysis

A statistician at Japanese Institute of Statistical Technology (Tokyo, Japan) analyzed all the data using SPSS Statistics 25 (IBM Corporation, Armonk, NY, USA). A Mann–Whitney U test was performed to determine significant between-group differences in the mental health scores. The significance level was set at *p* < 0.05.

## 3. Results

Valid data were acquired from 226 participants (88 males and 138 female), including 224 Japanese, 1 Chinese, and 1 Korean, with 98 in the pre-pandemic group (mean age: 44.16 years, standard deviation (SD): 18.28 years) and 128 in the during-pandemic group (mean age: 43.80 years, SD: 19.71 years) (Table 1). The proportion of female participants in the pre-pandemic group was slightly higher than that in the during-pandemic group, but there was no significant bias according to the Fisher’s exact test (*p* = 0.273). There was no difference between the pre- and during-pandemic groups in terms of gender, age, or diagnosis.

Overall, the GHQ-30 total, the GHQ-30 subscales (“social dysfunction”, and “anxiety and dysphoria”), and the STAI-S scores are significantly higher in the during-pandemic than in the pre-pandemic group (Table 2).

The GHQ-30, CES-D, and STAI scores according to sex and period (months) after the onset of the COVID-19 pandemic are also shown in Table 2. There were significant differences in the GHQ-30 total, the GHQ-30 subscales (“general illness”, “sleep disturbance”, and “anxiety and dysphoria”), and the STAI state scores between the pre- and during-pandemic groups of male participants, but no statistically significant differences in females were observed. Thus, the tendencies were different between males and females.

In addition to a significant increase in the total GHQ-30, “social dysfunction” and STAI-S scores in 10–30 year-olds was noted in the during-pandemic compared to the pre-pandemic group, “sleep disturbance” and CES-D scores also significantly increased. No significant differences can be observed in any items in the 40–50-year and ≥60year-old patients (Table 2).

As a result of each elapsed time after the pandemic, the STAI-S score showed a significant difference at 0–6 months after the Japanese government first declared a state of emergency compared to the pre-pandemic group. The GHQ-30 subscale (“social dysfunction” and “anxiety and dysphoria”) scores show a significant difference at 7–12 months after the Japanese government first declared a state of emergency compared to the pre-pandemic group (Table 3).

## 4. Discussion

### 4.1. Summary of Findings

This study examined the mental health status of patients visiting a psychosomatic internal medicine department for the first time before and during the COVID-19 pandemic in Japan.

As hypothesized, mental health status was worse in patients who visited the psychosomatic internal medicine department during the pandemic than in those who visited before the pandemic.

During the pandemic, social dysfunction and anxiety scales yielded higher scores among all participants. In addition, male participants yielded higher scores on general illness and sleep disturbance scales.

By the age groups, the scores of 10–30-year-olds on social dysfunction, sleep disturbance, anxiety, and depressive scale, increased during the pandemic.

Regarding the elapsed time after the pandemic, the anxiety scale yielded higher scores at 0–6 months, and social dysfunction and anxiety scales yielded higher scores at 7–12 months after the Japanese government first declared a state of emergency.

Japan is characterized by its “high-context” society, in which homogeneity, group harmony, and collectivism are valued, and people are expected to meet certain expectations and follow social norms [29]. Lee et al. [30] reported that individuals with a history of psychiatric disorders may be prone to the recurrence of psychiatric illnesses after the pandemic. Indeed, patients who have experienced mental health issues often experience depression and anxiety and are therefore considered most affected by the pandemic.

Based on the results, changes in mental health could be caused by several factors, such as the loss of social contacts, insecure situations, such as domestic infection due to the pandemic, limited school or university attendance, job insecurity or loss, and worries about parents’ health.

The recent state of health emergency resulted in social distancing from friends, colleagues, and relatives. Given the concerning daily news reports, the uncertain future, and exponential fear, the rising rate of anxiety, stress, and depression seems inevitable [3].

### 4.2. Age Difference of GHQ-30 and STAI

Overall, the GHQ-30 results indicate a large increase in “anxiety and dysphoria” due to “social dysfunction”. Social activity in the 10–30 age group declined and presented as a decrease in sleep quality. A previous Japanese study [31] identified the following major themes associated with pandemic distress for young people: (1) frustration with increased workload and struggles with stress relief; (2) concern about relatives; and (3) desire to achieve work satisfaction when dealing with infection control. In a collectivistic culture where people cannot violate in-group commitments regardless of stress, a collective universal behavioral campaign might cause excessive workloads. Another Japanese study reported [32] that severe mental distress and anxiety symptoms were more severe in younger than in older people. The existence of coping mechanisms contributed to the reduction of severe mental distress and anxiety symptoms. The results of this study may be related to these factors. However, no significant between-group differences were found in those aged ≥40 years, which may be due to the differences in these factors.

### 4.3. Gender Difference of GHQ-30

The GHQ-30 subscale (“general illness”, “sleep disturbance”, and “anxiety and dysphoria”) scores were significantly different between the males pre- and during-pandemic groups only. Hence, we conclude that these subscale results are meaningful. It is suggested that men are psychologically distressed. Cultural and social environments are different among males and females (e.g., life events, gender role norms at home, job positions, and/or salary gaps) [19]. Male stress responses may predominantly involve the traditional “fight and flight” reaction, whereas female stress responses may be better characterized by “tend and befriend”, involving nurturing activities and the creation of social networks [18]. Elhai et al. suggested that internet and cell phone use may be part of the coping strategies used in response to the distress secondary to the pandemic [33]. It is possible that females were less affected than males because they were able to form friendships online without having to meet in person. However, from the perspective of men, it is difficult to fight the virus, and in the global virus pandemic, it is thought that their stress increased since they were unable to escape.

In addition, women have various coping strategies and use them more frequently than men [34]. Women reportedly tend to focus on emotional coping [19] and mobilize greater social support during periods of stress [34]. In contrast, men prefer coping styles that can be practiced by themselves rather than relying on others. This could be why men rarely share their feelings under stress and are more likely to use alcohol, tobacco, or drugs than women [19]. Another reason for the sex difference in traditional Japanese women, Yamatonadeshiko, is their tendency to be patient, which is generally perceived as appealing by the society [29]. In Japan, such women may have acquired greater “patience” than they could apply during the COVID-19 pandemic. On the contrary, men in Japan financially support their families; the pandemic may have meant that they were unable to work, reducing their socioeconomic activity and lowering their income, which might have raised concerns about the future. Such a difference may also make a difference between men and women.

### 4.4. Period Difference of GHQ-30

Looking at the different periods after the outbreak, “social dysfunction” was not significant in the 0–6 months subgroup, but was statistically significant in the 7–12 months subgroup compared to the before-pandemic group. This indicates that “social dysfunction” became increasingly apparent more than 6 months after the Japanese government first declared a state of emergency.

These results support those of a report, which stated that recession led to socioeconomic inequalities based on age and gender during or after the lockdown, even after 6 months [35].

No significant between-group differences were found in the GHQ-30 somatization and depression scores. Regarding somatization, psychosomatic patients have difficulty noticing their physical condition, which is called alexithymia. This could explain why no significant changes in somatization were observed in the psychosomatic patients. Different results would be expected in patients diagnosed with emotional disorders, such as depression.

### 4.5. Overall STAI Scores

For the STAI test, state anxiety was significantly different between the two groups, but trait anxiety was not. While the STAI-S assesses responses at specific moments in time, the STAI-T determines how participants generally feel [23]. The finding that trait anxiety did not change as a result of the COVID-19 pandemic could indicate the characteristics of the STAI-T, which determines how participants generally feel, and is relatively stable regardless of the situation [23].

### 4.6. Limitations

This study has quite a few limitations. First, this was a single-center study conducted at a university hospital. A single center cannot represent the situation of the whole country. For future research, it is recommended that all psychosomatic internal medicine clinics implement all types of the selected psychological tests. Second, this study used a self-reported standardized questionnaire, and this survey was conducted during our daily practice, considering the burden on patients. Although the GHQ-30 and the STAI are not highly sensitive to changes that may be occurring at the contextual level, the questionnaire used in this study was the one used in previous studies (e.g., [36,37] on changes in mental health due to the COVID-19 pandemic). It is considered effective to evaluate mental health and research from various aspects, including a quantitative study, in the future to obtain more detailed information.

Regarding future research, it is necessary to make corrections by multivariate analysis, such as logistic regression analysis, or the adjustment of the significance level by multiple comparison, when confirming the association with external factors, such as the presence or absence of mental illness or other factors. If the effects of COVID-19 also affect social activities, it will be important to understand how people’s anxiety and depression change. The results of this study concerning the period following the pandemic indicate that social activity and anxiety are associated, such that an increase in social activities reduces the observed depression and anxiety. In the meantime, it is necessary to track how their mental health changes over time.

## 5. Conclusions

The COVID-19 pandemic affects the mental health of patients, as noted by the patients visiting the psychosomatic medicine clinic. Gender difference was noted, especially with males being more vulnerable to such disorders than females. Additionally, by age group, participants aged 10–30 years were affected the most, more than half a year after the onset of the pandemic. Patients visiting the outpatient department of psychosomatic medicine may be affected by the pandemic, but there are differences depending on gender, age, and time from the onset of the pandemic. It may be useful to support the patients in line with these differences. Future studies, including qualitative studies, are required to uncover why the patients who visit the department of psychosomatic internal medicine are more affected, and to develop effective approaches to improve their mental health.

## Figures and Tables

**Table 1 ijerph-19-02488-t001:** Sex, the mean age ± SD, and age of the participants in the pre-pandemic and during-pandemic groups (top). Subgroups divided to “pre-pandemic subgroup”, “0–6 months subgroup”, and “7–12 months subgroup (bottom).

		Gender		Mean Age ± SD		Age	
		Females	Males		10–30s	40–50s	60–83s
Group	**Pre-pandemic**	64	34	44.16 ± 18.28	43	33	22
		65.3%	34.7%		43.9%	33.7%	22.4%
	**During-pandemic**	74	54	43.80 ± 19.71	57	37	34
		57.8%	42.2%		44.5%	28.9%	26.6%
**Total**		138	88		100	70	56
		61.1%	38.9%		44.2%	31.0%	24.8%
		**Pre-Pandemic**	**0** **–6 Months Subgroup**		**7–12 Months Subgroup**		**Total**
** *n* **		98	58		70		226
		43.4%	25.6%		31.0%		100%

The 0–6 months subgroup: 0–6 months after the Japanese government first declared a state of emergency; 7–12 months subgroup: 7–12 months after the Japanese government first declared a state of emergency.

**Table 2 ijerph-19-02488-t002:** GHQ, CES-D, and STAI score comparisons between pre-pandemic and during-pandemic groups, based on the total scores, sex, and age groups.

	**Total**					**Male**					**Female**				
	* **n** * **, Mean ± SD**					* **n** * **, Mean ± SD**					* **n** * **, Mean ± SD**				
	* **n** *	**Pre-Pandemic**	**N**	**During-Pandemic**	***p*-Value**	* **n** *	**Pre-Pandemic**	**N**	**During-Pandemic**	***p*-Value**	** *n* **	**Pre-Pandemic**	**N**	**During-Pandemic**	***p*-Value**
GHQ total score	92	12.98 ± 8.50	125	15.15 ± 7.03	0.023 *	31	10.58 ± 8.12	52	15.56 ± 6.57	0.005 *	61	14.20 ± 8.49	73	14.86 ± 7.37	0.38
General illness	92	2.65 ± 1.73	125	3.06 ± 1.45	0.11	31	2.16 ± 1.53	52	3.23 ± 1.52	0.003 *	61	2.90 ± 1.79	73	2.95 ± 1.40	0.80
Somatic symptom	92	2.26 ± 1.81	125	2.29 ± 1.51	0.85	31	1.90 ± 1.92	52	2.52 ± 1.55	0.095	61	2.44 ± 1.74	73	2.12 ± 1.47	0.28
Sleep disturbance	92	2.57 ± 1.72	125	2.99 ± 1.69	0.073	31	1.84 ± 1.66	52	3.13 ± 1.63	0.001 *	61	2.93 ± 1.65	73	2.89 ± 1.73	0.89
Social dysfunction	92	1.83 ± 1.70	125	2.31 ± 1.68	0.034 *	31	1.48 ± 1.53	52	2.17 ± 1.64	0.058	61	2.00 ± 1.77	73	2.41 ± 1.72	0.17
Anxiety and dysphoria	92	2.26 ± 1.96	125	2.86 ± 1.67	0.028 *	31	1.87 ± 2.01	52	2.85 ± 1.59	0.029 *	61	2.46 ± 1.91	73	2.86 ± 1.74	0.23
Suicidal depression	92	1.35 ± 1.80	125	1.59 ± 1.93	0.40	31	1.29 ± 1.72	52	1.54 ± 1.90	0.64	61	1.38 ± 1.85	73	1.63 ± 1.95	0.49
CES-D	94	21.06 ± 12.73	128	23.91 ± 12.57	0.076	33	19.09 ± 10.82	54	24.31 ± 12.05	0.051	61	22.13 ± 13.61	74	23.61 ± 13.01	0.51
STAI trait	96	51.53 ± 13.70	127	53.31 ± 12.79	0.24	33	51.67 ± 12.43	54	54.69 ± 12.46	0.21	63	51.46 ± 14.41	73	52.30 ± 13.02	0.67
STAI state	97	51.49 ± 12.42	127	54.59 ± 11.29	0.033 *	33	49.30 ± 12.03	54	55.61 ± 11.11	0.015 *	64	52.62 ± 12.56	73	53.84 ± 11.44	0.41
	**10–30s**					**40–50s**					**60–83s**				
	** *n* ** **, Mean ± SD**					** *n* ** **, Mean ± SD**					** *n* ** **, Mean ± SD**				
	** *n* **	**Pre-Pandemic**	**N**	**During-Pandemic**	** *p* ** **-Value**	** *n* **	**Pre-Pandemic**	**N**	**During-Pandemic**	** *p* ** **-Value**	** *n* **	**Pre-Pandemic**	**N**	**During-Pandemic**	** *p* ** **-Value**
GHQ total score	40	13.33 ± 8.18	57	16.60 ± 6.16	0.032 *	31	15.26 ± 9.95	37	15.38 ± 6.98	0.92	21	8.95 ± 4.92	31	12.23 ± 7.89	0.17
General illness	40	2.55 ± 1.66	57	3.16 ± 1.37	0.081	31	2.94 ± 1.98	37	3.16 ± 1.42	0.90	21	2.43 ± 1.47	31	2.77 ± 1.63	0.39
Somatic symptom	40	2.47 ± 1.87	57	2.56 ± 1.46	0.80	31	2.55 ± 1.88	37	2.62 ± 1.40	0.92	21	1.43 ± 1.36	31	1.39 ± 1.41	0.86
Sleep disturbance	40	2.28 ± 1.52	57	3.00 ± 1.71	0.035 *	31	2.90 ± 1.92	37	3.11 ± 1.60	0.80	21	2.62 ± 1.77	31	2.84 ± 1.79	0.71
Social dysfunction	40	1.90 ± 1.63	57	2.54 ± 1.55	0.037 *	31	2.26 ± 1.90	37	2.19 ± 1.73	0.96	21	1.05 ± 1.28	31	2.03 ± 1.85	0.070
Anxiety and dysphoria	40	2.48 ± 1.99	57	3.21 ± 1.56	0.090	31	2.74 ± 1.98	37	2.92 ± 1.53	0.84	21	1.14 ± 1.42	31	2.13 ± 1.84	0.065
Suicidal depression	40	1.50 ± 1.85	57	2.02 ± 2.03	0.18	31	1.87 ± 2.00	37	1.38 ± 1.80	0.30	21	0.29 ± 0.64	31	1.06 ± 1.75	0.17
CES-D	42	21.33 ± 13.31	57	26.70 ± 11.38	0.021 *	30	23.87 ± 13.55	37	23.05 ± 13.71	0.77	22	16.73 ± 9.33	34	20.15 ± 12.41	0.38
STAI trait	42	51.98 ± 13.89	56	56.07 ± 11.70	0.088	32	55.34 ± 13.92	37	53.43 ± 12.08	0.54	22	45.14 ± 10.98	34	48.65 ± 14.20	0.26
STAI state	43	51.91 ± 11.68	56	57.16 ± 9.92	0.015 *	32	54.72 ± 13.05	37	54.95 ± 11.13	0.89	22	46.00 ± 11.54	34	49.97 ± 12.43	0.25

GHQ: General Mental Health Scale; CES-D: Center for Epidemiologic Studies Depression Scale; STAI: State–Trait Anxiety Inventory. * *p* < 0.05.

**Table 3 ijerph-19-02488-t003:** GHQ, CES-D, and STAI score comparisons between periods in the during-pandemic group.

	vs. 0–6 Months Subgroup					vs. 7–12 Months Subgroup				
	*n*, Mean ± SD					*n*, Mean ± SD				
	*n*	Pre-Pandemic	*n*	During-Pandemic	*p*-Value	*n*	Pre-Pandemic	N	During-Pandemic	*p*-Value
GHQ total score	92	12.98 ± 8.50	58	15.29 ± 7.57	0.056	92	12.98 ± 8.50	67	15.03 ± 6.59	0.058
General illness	92	2.65 ± 1.73	58	3.03 ± 1.56	0.20	92	2.65 ± 1.73	67	3.09 ± 1.37	0.16
Somatic symptom	92	2.26 ± 1.81	58	2.48 ± 1.50	0.40	92	2.26 ± 1.81	67	2.12 ± 1.51	0.64
Sleep disturbance	92	2.57 ± 1.72	58	3.02 ± 1.78	0.11	92	2.57 ± 1.72	67	2.97 ± 1.61	0.15
Social dysfunction	92	1.83 ± 1.70	58	2.16 ± 1.71	0.24	92	1.83 ± 1.70	67	2.45 ± 1.66	0.020 *
Anxiety and dysphoria	92	2.26 ± 1.96	58	2.86 ± 1.86	0.090	92	2.26 ± 1.96	67	2.85 ± 1.50	0.048 *
Suicidal depression	92	1.35 ± 1.80	58	1.62 ± 1.92	0.45	92	1.35 ± 1.80	67	1.57 ± 1.95	0.50
CES-D	94	21.06 ± 12.73	58	23.10 ± 12.89	0.31	94	21.06 ± 12.73	70	24.57 ± 12.34	0.058
STAI trait	96	51.53 ± 13.70	58	52.95 ± 12.85	0.42	96	51.53 ± 13.70	69	53.62 ± 12.82	0.26
STAI state	97	51.49 ± 12.42	58	55.17 ± 11.81	0.048 *	97	51.49 ± 12.42	69	54.10 ± 10.91	0.11

GHQ: General Mental Health Scale; CES-D: Center for Epidemiologic Studies Depression Scale; STAI: State–Trait Anxiety Inventory * *p* < 0.05. 0–6 months subgroup: 0–6 months after the Japanese government first declared a state of emergency; 7–12 months subgroup: 7–12 months after the Japanese government first declared a state of emergency.

## Data Availability

The datasets presented in this study are available on request from the corresponding author.

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
