# Peer review of "Mental Health Status before and during the COVID-19 Pandemic in Patients First Visiting a Psychosomatic Internal Medicine Clinic in Tokyo, Japan"

_ijerph, 2022, doi:10.3390/ijerph19042488_

Round 1
Reviewer 1 Report
In this paper, Fumio Shaku and colleagues examined the psychological effects of the pandemic in patients first visiting a psychosomatic internal medicine clinic in Japan.
There are some major issues that I wish the authors would fix to improve the value of the article.
- Are the two groups comparable? I suggest better explaining the characteristics of the pre-pandemic and the during-pandemic group to understand if the two groups are comparable (not only by sex and mean age but also, e.g., by diagnosis).
- In my opinion if the authors desire to make an important contribution to the topic, they need to enrich the conclusions and better explain the clinical or epidemiology utility of their findings.
Reviewer 2 Report
The methodology and questionnaires used are good, but in some places the comprehensibility and transparency could be improved. The following points should be noted:
-
line 67: the previously mentioned studies do not include evidence of increased susceptibility for Covid 19 in psychosomatic patients. Maybe you rather mean an increased vulnerability for the psychological consequences of Covid?
-
Participants: What was the average length of time subjects were in the clinic? What psychosomatic disorders did the patients have? Where is the relation to psychosomatic issues and why was this particular patient group selected? Please clarify.
-
Discussion: Increased anxiety due to social media use in the 10-30 age group seems a bit out of the air, social media (including news) use was not even considered in this study, was it? Increased anxiety could therefore be due to many factors, e.g. loss of social contacts, insecure situation due to the pandemic, limited school or university attendance, job insecurity or loss, worries about parents’ health etc. In my opinion, the gender difference also requires a more differentiated discussion.
Reviewer 3 Report
Thank you for the opportunity to review this paper. This study studies the mental health status before and during the COVID-19 pandemic. Many studies reported that mental health problems are seen during the COVID-19 pandemic. Please discuss clearly the gap and the significance of the study.
In 2.3.3., the description of CES-D is too brief. What are the scoring and the interpretation of the CES-D?
The results are unclear. The descriptive statistics have not been reported, only reported the p-valve to indicate the result's significant or not. It cannot reflect the mental health status of the patients before and during the COVID-19 pandemic in Japan.
Reviewer 4 Report
The General Mental Health Scale (GHQ), the State-Trait Anxiety Inventory (STAI), and the Center for Epidemiologic Studies Depression Scale (CES-D) were used to assess mental health, state and trait anxiety, and depression, respectively. Therefore, I think the manuscript is methodologically sound. However, I believe the paper can be strengthened furthermore if the study addresses three issues as follows:
1.The abstract suggests a structured format: objective, method, results and conclusion.
2.this was a single-center study conducted at a university hospital.Therefore, it is recommended to add specific city or region names in the title of the article, rather than just the name of the country. A single city or hospital cannot represent the situation of the whole country, which is a fatal defect.
3.In the discussion section, it is suggested to add several subheadings according to the content, so that the hierarchy is more clear and the meaning can be expressed more clearly.
4.In the first sentence of the discussion section, "The first study" is not very good, I suggest changing it to another expression.
Reviewer 5 Report
This is a study that deals with a current issue such as mental health effects due to the Covid-19 pandemic, and precisely because of this, the objective of this study does not seem very innovative. However, the novelty lies in the fact that it deals with a psychosomatic medicine service, although this is not very well contextualized in the study. Despite its topicality, the study shows shortcomings which in my opinion should be improved in order to be published.
Summary: The topic is well presented but the conclusion it reflects does not match the reality of the work they do and the results they show.
Introduction: The information in the introduction is written in a general way and is not focused on the content of the study. For example, it refers to a study of mental health in university students, but this is not the objective of the study, nor is this specific group used to obtain the sample. Furthermore, reading their discussion, they allude on several occasions to the specific characteristics of Japanese society, even indicating gender differences in terms of social behaviour, so the introduction should contextualize the population more, due to the great importance this has for justifying their results.
Material and methods: In the section on study design, the study design is missing. Is it a quantitative study? Descriptive? Experimental? This should be included. As for the sample, the only inclusion criterion was that they were first-time patients and over the age of 18? No consideration was given to whether or not they had any previous mental health or psychosomatic problems, whether or not they had been to any other institution for therapy? Were all participants Japanese or were there any participants of other nationalities? There should be more specific inclusion and exclusion criteria, otherwise the sample will be very biased. There is also insufficient socio-demographic data to understand the sample.
How was the sample recruited and who did the outreach and recruitment? Once they were recruited, who explained the study to them? According to your study, your government does not require informed consent for this data collection, but were they given an information sheet? This is necessary, as it would verify that the participants have understood the object of the study and what participation in the study entails, and that they are given the possibility to resolve possible doubts. From what I gather from your text, the questionnaire was self-reported, how can you know that the participants have correctly understood what is being asked about? This raises doubts for me, as there are very complex items that might need some clarification from the researchers.
As for the questionnaires used, I do not think that they are the most suitable for measuring the data that have been proposed as objectives. For example, the GHQ-30 and the STAI are not very sensitive to changes that may be occurring at the contextual level, so they are not good instruments for measuring this affect at the individual level.
The sample is too small for the data analysis to be considered as significant, 119 Group A and 141 Group B. Furthermore, there is no homogeneity of the participants, and therefore they are not likely to be able to make a comparison between them that would yield significant results.
Discussion: In general, the discussion deals with the presentation of data that have nothing to do with the results obtained, perhaps because the sample is not adequate and it is difficult to justify the results themselves. They make a very explicit reference to young people between 10-30 years of age, but in their sample we do not know how many participants were in this range, nor how they dealt with new technologies and social networks, in fact these data are not collected or measured in the study. You also refer to the influence of the media, but how has this been included in the study? How has this influence been measured? Moreover, there is no discussion of this issue in the introduction, which may mislead the reader. In your results it appears that men seem to be more affected in their mental health during the pandemic compared to women, but this difference is not understood because there is no correct contextualisation of Japanese society. You even compare your results with an Australian study and from what you state, Australian social behaviour has nothing to do with Japanese social behaviour. It would need further justification. However, you only make a very brief reference to this. In the limitations section, you mention as a bias that they were patients attending psychosomatic consultation, but I believe that this is precisely the aim of the study, so it would not be a limitation.
Round 2
Reviewer 1 Report
I confirm that my comments and concerns have been addressed by the authors.
Author Response
Thank you for your confirmation message.
Reviewer 3 Report
Thank you for the opportunity to review this paper.
In the introduction, more explanations are needed in some areas, such as occupational differences (in line 53) and cultural differences (in line 71).
In the result parts, there are some errors in the article and the table, such as data in line 153 and the data in table 1. The inconsistency of the data. In Tables 2 & 3, are there any missing data from the subjects? Data cleaning is required prior to data analysis. Suggest the author to review the data analysis session.
In the discussion part, the summary of findings is too brief and needs more explanation.
